# Impact of High-Speed Rail Construction on the Environmental Sustainability of China's Three Major Urban Agglomerations

Sining Zhu, Zhou Zhou *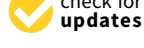, Ran Li * and Wenxing Li

School of Economics and Management, Beijing Jiaotong University, Beijing 100044, China;
14113086@bjtu.edu.cn (S.Z.); wxli@bjtu.edu.cn (W.L.)
* Correspondence: 20113001@bjtu.edu.cn (Z.Z.); li.ran@bjtu.edu.cn (R.L.)

**Abstract:** Under the background of global warming, it is of great significance to explore how to realize environmentally sustainable development. This paper takes China's three major urban agglomerations as the research objects: Yangtze River Delta, Beijing–Tianjin–Hebei, and Pearl River Delta. Generally, we use carbon emission efficiency to represent the sustainable development of the environment. Then we use the city-level panel data of the three urban agglomerations from 2006 to 2019 to construct the slacks-based measure integrating data envelopment (SBM-DEA) model for calculating each city's carbon dioxide emission efficiency. Finally, we construct the spatial difference-in-differences (SDID) model to explore the impact of high-speed rail construction on each urban agglomeration's carbon dioxide emission efficiency and its internal mechanism. The findings are as follows: (1) On the whole, high-speed rail construction improves urban agglomerations' carbon dioxide emission efficiency. Meanwhile, it has a positive spatial spillover effect on surrounding areas. (2) In terms of urban agglomerations, high-speed rail construction has significantly promoted carbon emission efficiency in the Beijing–Tianjin–Hebei region. However, it has had negative external effects on the surrounding areas. (3) From the perspective of mechanism analysis, the construction of high-speed rail has promoted manufacturing agglomeration in the Pearl River Delta region and, at the same time, has had a negative impact on the local carbon dioxide emission efficiency. This study has strong policy implications for promoting the sustainable development of the three major urban agglomerations.

**Keywords:** high-speed rail; three major urban agglomerations; environmental sustainability; SBM-DEA model; SDID model

## 1. Introduction

Under the background of global climate and environmental change, the scarcity of environment and natural resources has increased, and economic activity restrictions have been continuously enhanced. Environmental problems have become the main challenge restricting sustainable development, especially in developing countries [1]. As the largest developing country, China plays a vital role in promoting the sustainable development of the global environment [2]. In September 2020, in the 75th U.N. General Assembly, President Xi Jinping made a solemn commitment to the world that China's carbon dioxide emissions will peak by 2030 and be carbon neutral by 2060. In 2010, the transportation sector produced about 23% of energy-related carbon dioxide emissions. If we do not take any positive mitigation measures, the carbon dioxide emissions related to transportation may double by 2050 and even triple by 2100 [3]. Therefore, the transportation sector has great potential to achieve the goal of carbon neutrality and carbon peaking [4]. At present, high-speed rail construction is considered a fundamental approach to promoting the environment's sustainable development [5]. On the one hand, high-speed rail generates less carbon dioxide emissions than civil aviation, automobile, and other transportation modes [6,7]. On the other hand, large-scale high-speed rail construction will significantly

reduce transportation costs and promote the flow of talent, capital, and technology, thus promoting regional technological innovation and reducing the environmental pollution of enterprise production [8,9].

At present, China's economic development focuses on developing the construction of urban agglomerations with big cities as the core. Transportation development is an essential means of promoting cluster growth, and strengthening the organic combination of transportation and urban agglomeration construction is particularly important for backward areas. Beijing–Tianjin–Hebei, Yangtze River Delta, and Pearl River Delta are the most critical locations for China to implement the strategic layout of urbanization. Promoting the deep integration of these three major urban agglomerations and giving full play to their primary functions and potential are the key points of China's regional optimization and coordinated development.

During the 13th Five-Year Plan period, the rail transit development of the three major urban agglomerations achieved many successes. The operating mileage of the Beijing–Tianjin–Hebei high-speed railway increased 41.6% from 1616.3 km to 2288.6 km. The high-speed rail mileage in the Yangtze River Delta increased 84.9% from 3250 km to 6008 km. Meanwhile, the high-speed rail mileage in Guangdong–Hong Kong–Macao Greater Bay Area was 1232 km at the end of 2019. Its high-speed rail network density was the highest among the three major urban agglomerations. Overall, the current high-speed rail mileage in Beijing–Tianjin–Hebei, Yangtze River Delta, and Guangdong-Hong Kong-Macao Greater Bay Area has exceeded 9500 km, accounting for a quarter of the national total. In addition, in the outline of the 14th Five-Year Plan, the goal of these three urban agglomerations is mentioned: "Build Beijing–Tianjin–Hebei in orbit; accelerate the construction of inter-city railway in Guangdong–Hong Kong–Macao Greater Bay Area; achieve complete coverage of high-speed rail in cities above the prefecture-level in the Yangtze River Delta". The high-speed rail construction has dramatically shortened the distance between central marginal towns and cities in the three major urban agglomerations. It will promote the spatial correlation of urban agglomerations. So, in recent years, has the rapid development of high-speed rail construction in China's three major urban agglomerations promoted carbon dioxide emission efficiency? If not, what is the limiting factor, and how can the environmental effect of high-speed rail be exerted in the future?

Recently, academic circles have widely focused on the relationship between high-speed rail construction and environmental pollution. Moreover, many scholars have conducted empirical research on high-speed rail construction and have found that it significantly improved the environmental quality.

Firstly, some researchers found that high-speed rail is cleaner than other modes of transportation. The continuous development of high-speed rail construction replaces much highway and civil aviation passenger traffic, thus reducing carbon dioxide emissions [10–12]. Secondly, high-speed rail construction has dramatically reduced transportation costs, bringing industrial agglomeration [8]. It will benefit the upgrading of regional industrial structures and further reduce the emission of environmental pollutants [9,13,14].

Thirdly, the construction of high-speed rail is also conducive to the flow of labor, capital, technology, and other factors. On the one hand, with the weakening of the barriers to factor flow, factor allocation efficiency has been dramatically improved, thus reducing the emission intensity of industrial pollution [9,15]. On the other hand, reducing transportation costs is conducive to the technical connection between regions and promotes regional technological innovation. Improving production efficiency and cleaner production technology will be conducive to energy saving and emission reduction [16,17]. In addition, the construction of high-speed rail has also strengthened the interregional ties, which will impact the local environment and have a spatial spillover effect on the surrounding areas [18]. Fang [14] found that high-speed rail construction can reduce smog pollution by upgrading industrial structures and developing the real estate market. At the same time, urban smog pollution has a strong spatial correlation. Li [19] also found that the environmental pollution of enterprises in a region will be transferred to the surrounding

areas through the high-speed rail network. Further, the spillover effect of pollution space is significantly related to the economic development, environmental investment, the number of enterprises, and other factors in the region.

However, some scholars believe that the construction of high-speed rail will have a negative impact on the environment. On the one hand, as the opening of high-speed rail greatly facilitates residents' travel, it will lead to more traffic demand to a certain extent, which will lead to an increase in carbon dioxide emissions [20–23]. Givoni and Dobruszkes [20] found that the opening of a high-speed railway leads to a 20% travel demand increase. On the other hand, the construction of high-speed rail will generate a large number of pollutants [2,24]. Yue [25] comprehensively considered the environmental effects during the construction of the Beijing–Shanghai railway line and pointed out that the process included greenhouse gas emissions and PM2.5 emissions, fossil resource consumption, surface water eutrophication, and other issues. Kaewunruen [26] found that 64.86% of carbon dioxide emissions and 54.31% of energy consumption in the whole life cycle of high-speed rail construction come from the construction stage.

The literature has discussed the relationship between high-speed rail construction and environmental pollution and the possible intermediate mechanism from different perspectives. However, the existing research still has the following problems: (1) Many pieces of literature have discussed the impact of high-speed rail construction on carbon dioxide emissions and PM2.5, but little research pays attention to its effects on environmental efficiency compared with the total amount of pollutants. The carbon dioxide emission efficiency can better reflect whether a region has green and low-carbon development potential and can achieve sustainable development [27]. (2) Most literature builds a difference-in-difference model to compare the difference in carbon dioxide emissions before and after opening a high-speed rail. However few pieces of literature consider the spatial correlation brought by high-speed rail construction. However, if the spatial factor is not considered, the estimation result will be biased. (3) At present, most literature is based on the data of all cities in China. However, different regions in China have different development characteristics and resource endowments, and the impact of high-speed rail construction on the environment of different areas should be different. Therefore, this paper intends to use the data of three major urban agglomerations in China to build the SBM-DEA and spatial DID models to explore the relationship and internal mechanism between high-speed rail construction and environmentally sustainable development.

The research contribution of this paper is mainly manifested in the following three aspects. Firstly, we built the SBM-DEA model to measure each city's carbon dioxide emission efficiency in the three urban agglomerations in order to measure the environmental sustainability of urban agglomerations more accurately. Secondly, we constructed a spatial DID model to estimate the impact of high-speed rail construction on carbon dioxide emission efficiency more accurately by adding spatial factors into the model. Thirdly, we provided corresponding policy suggestions for regions with different development characteristics by comparing the impacts of high-speed rail construction on the environmental efficiency of three major urban agglomerations.

The remainder of this study is structured as follows. Section 2 explores the relationship between high-speed rail construction and sustainable development of urban agglomeration environments through theoretical mechanism analysis. Section 3 describes the SBM-DEA model used to calculate three major urban agglomerations' carbon dioxide emission efficiency. In Section 4, we introduce the model construction process and the index selection. Section 5 mainly presents regression results, robustness analysis, mechanism test, etc. The last section is the conclusion and policy implications.

## 2. Theoretical Analysis and Research Hypothesis

### 2.1. High-Speed Rail Construction and Environmental Sustainability

Generally, compared with air, highway, and other modes of transportation, high-speed rail is considered the cleanest mode of transportation [6,7]. With the vigorous development

of high-speed rail construction and because of its fast and convenient characteristics, a large number of passengers have turned from aviation and highway to railway travel, thus reducing the overall carbon emissions [10]. According to relevant statistics, from 2008 to 2016, China's high-speed rail network eliminated 14.76 million tons of carbon dioxide by decreasing road passenger and cargo traffic [28]. In addition, the construction of high-speed rail has dramatically reduced the inter-regional transportation cost and exacted some effects on the environment indirectly. Firstly, reduced time cost will stimulate the movement of talent, contributing to the region's increased human capital and innovation potential [29]. Secondly, high-speed rail will also promote the free flow of capital, which provides the region's financial support to upgrade the regional industrial structure [30]. Thirdly, improved accessibility will lead to technology transfer, which is beneficial for the improvement of production capabilities [31]. With the increased innovation potential and optimal factor allocation, the production efficiency will be improved and create less pollution.

The continuous improvement of the high-speed rail network has greatly facilitated residents' travel, leading to more traffic demand, leading to increased carbon dioxide emissions [20]. In addition, large-scale high-speed rail construction will also have significant negative impact on the environment [24]. Overall, the negative effects mainly include two aspects: first, a large amount of carbon dioxide will be generated in the process of manufacturing high-speed rail cars; second, during the construction process, a large number of pollutants, such as wastewater, smoke dust, and waste gas will be produced [4]. Therefore, the transportation derived demand and the negative externalities in the construction process will significantly restrict the environmental sustainability of development. This paper puts forward Hypothesis 1 (H1) based on analyzing the positive and negative external effects of high-speed rail construction on the environment.

**Hypothesis 1 (H1).** *When high-speed rail construction has more negative effects on the environment than positive ones, it is not conducive to the sustainable development of the urban agglomeration environment, and vice versa.*

### 2.2. High-Speed Rail Construction, Spatial Correlation, and Environmental Sustainability

The construction of high-speed rail has dramatically reduced transportation cost, extensively promoting the spatial correlation of regions [32], and has a related impact on the local environment.

On the one hand, the construction of high-speed rail is helpful to enhance the inter-regional correlation effect in urban agglomerations by promoting the flow of talent, capital, and technology. Then the advanced technologies, ideas, and other elements of central cities can much more easily flow into peripheral cities [29]. The technology spillover effect of the central city to the peripheral cities will help the region improve innovation, thereby reducing pollution and promoting environmental sustainability [18,33].

On the other hand, the reduction of transportation costs will also lead to many factors flowing from peripheral cities to central cities, which will lead to a considerable loss of production factors in backward cities [34,35]. Thus, these peripheral regions cannot carry out technological innovation, which inhibits the sustainable development of the region's environment. What's more, the construction of high-speed rail will also facilitate the migration of polluting enterprises from central cities to peripheral cities, which will lead to a significant decline in environmental sustainability in this region [8]. And then we proposed the Hypothesis 2 (H2) and Hypothesis 3 (H3).

**Hypothesis 2 (H2).** *High-speed rail construction can play a leading role in transportation from central cities to surrounding cities, thus producing a positive spatial spillover effect on environmental sustainability.*

**Hypothesis 3 (H3).** *High-speed rail construction promotes the migration of polluting enterprises to the peripheral areas, thus producing a negative spatial spillover effect on environmental sustainability in the peripheral regions.*

*2.3. High-Speed Rail Construction, Manufacturing Agglomeration, and Environmental Sustainability*

The manufacturing industry is the second-largest carbon emission source in China. The success of its emission reduction is the key to China's peak carbon dioxide emissions and carbon neutrality goals [36]. Generally speaking, the improvement of traffic accessibility has an essential impact on the location choice of manufacturing enterprises [37]. The construction of high-speed rail significantly changes the location conditions of the city by saving travel time. Furthermore, it affects the agglomeration of manufacturing industries in different cities [38].

On the one hand, industrial agglomeration will produce a scale effect, which will reduce its production cost. Then it will be more conducive to technological innovation, which promotes clean production and ultimately helps to promote the efficiency of carbon dioxide emission [18,33,39,40]. On the other hand, high-speed rail construction can enhance the agglomeration advantages of labor and other factors in backward areas. At the same time, the area has a lower environmental threshold, which helps to promote the local agglomeration of polluting manufacturing industries [8]. It then has a negative impact on its carbon dioxide emission efficiency. Based on the analysis, we proposed the Hypothesis 4 (H4). The relationship between HSR construction and environmental sustainability can be seen in Figure 1.

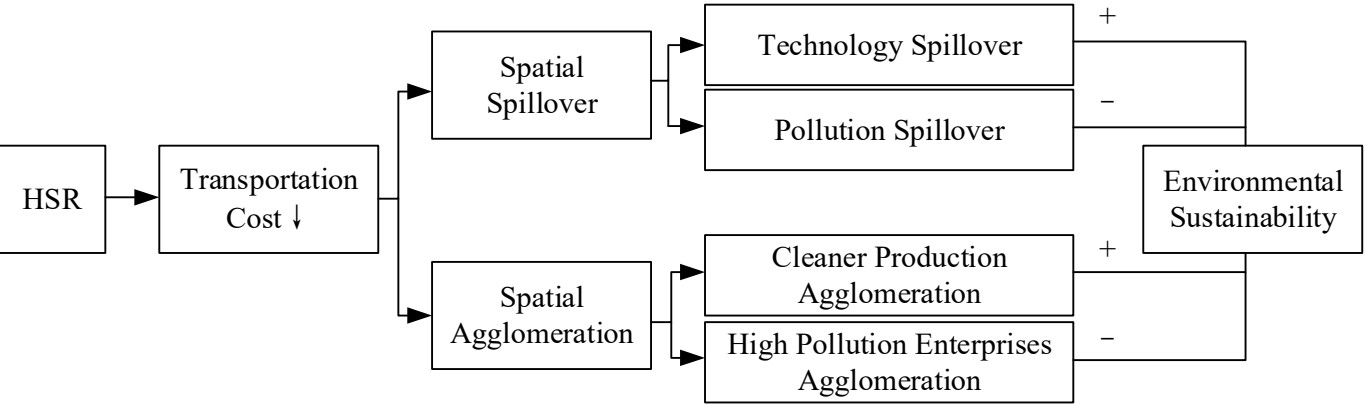

**Figure 1.** The logic roadmap of the relationship between HSR and environmental sustainability.

**Hypothesis 4 (H4).** *High-speed rail construction affects environmental sustainability by promoting manufacturing industries' agglomeration.*

## 3. Environmental Sustainability of Three Major Urban Agglomerations
*3.1. Definition of Environmental Sustainability*

Generally speaking, environmental sustainability is defined as protecting and enhancing environmental systems' production and regeneration capacity [41]. Higher carbon emissions severely restrict the environmental carrying capacity [42]. As the largest developing country, China plays a vital role in promoting the sustainable development of the global environment [2]. At present, China has launched two goals of "carbon neutrality" and "carbon peaking". Thus, reducing carbon emissions has become the focus of promoting high-quality economic development in China. The key to reducing carbon emissions is to improve carbon emission efficiency [27]. Carbon emission efficiency refers to the proportional relationship between production output (including desirable economic output and undesired carbon emissions) and production factor inputs under certain technological progress conditions. So it represents the resource allocation efficiency,

which reflects a region's environmentally sustainable development [43]. Therefore, this paper chooses carbon dioxide emission efficiency as a proxy variable for environmentally sustainable development.

### 3.2. Measurement of Carbon Dioxide Emission Efficiency

3.2.1. SBM-DEA Model Construction

Generally, we used the following methods to measure carbon emission efficiency: stochastic frontier analysis (SFA) and data envelopment analysis (DEA). The DEA method does not need to consider the functional relationship between input and output indicators. It does not need to assume the relationship between variables in advance, which avoids the influence of human subjective factors to a certain extent. Therefore, it has apparent advantages in measuring multi-input and output decision-making units' efficiency and is widely used in efficiency evaluation research.

DEA is based on the concept of Pareto optimal solutions and utilizes linear programming to evaluate the relative efficiency of decision-making units (DMUs). Charnes et al. [44] proposed a Charnes, Cooper and Rhodes (CCR) model based on fixed returns to scale. Subsequently, Banker et al. [45] revised the assumption of constant returns to scale (CRS) in the CCR model to variable returns to scale and proposed the Banker, Charnes and Cooper (BCC) model. Since the above models cannot solve the problems of undesired output and slack variables, Tone [46] proposed a non-radial and non-directed SBM (slack-based measure) model to evaluate each decision-making unit's efficiency better. This paper adopts the SBM model, considering undesired output to measure the static level of carbon dioxide emission efficiency in the three major urban agglomerations.

We have considered $N$ kinds of input ($x$), $M$ varieties of expected output ($y$), and $I$ kinds of unexpected outputs ($b$). Then, we constructed the following linear programming problem:

$$Min\theta = \frac{1 - \frac{1}{N}\sum_{n=1}^{N}\frac{s_n^x}{x_{n0}}}{1 + \frac{1}{M+I}\left(\sum_{m=1}^{M}\frac{s_m^y}{y_{m0}} + \sum_{i=1}^{I}\frac{s_i^b}{b_{i0}}\right)}$$

$$s.t.\begin{cases} \sum_{k=1}^{K}z_k x_{nk} + s_n^x = x_{n0}, n = 1, 2, \cdots, N \\ \sum_{k=1}^{K}z_k y_{mk} - s_m^y = y_{m0}, m = 1, 2, \cdots, M \\ \sum_{k=1}^{K}z_k b_{ik} + s_i^u = b_{i0}, i = 1, 2, \cdots, I \\ \sum_{k=1}^{K}z_k = 1 \\ s_m^y \geq 0, s_n^x \geq 0, s_i^b \geq 0, z_k \geq 0 \end{cases} \quad (1)$$

where $\theta$ is the carbon emission efficiency value to be measured, and the value range is $[0, 1]$, $x_{n0}$ represents the input vector of the $n$-th DMU, $y_{m0}$ means the $m$-th expected output, $b_{i0}$ indicates the $i$-th undesirable output, $z_k$ is the weight coefficient of input and output indexes, and $s_n^x$, $s_m^y$, and $s_i^b$ are the slack of input factors, desirable output, and undesirable output, respectively. When $\theta = 1$, $s_n^x = 0$, $s_m^y = 0$ and $s_i^u = 0$ of one DMU, the DMU is effective. When $\theta < 1$ of one DMU, the DMU is ineffective. The objective is to minimize $\theta$. The constraints mean the DMU can be improved by reducing inputs and undesirable outputs and increasing desirable outputs.

3.2.2. Input-Output Variables

Based on the research of Li [47], this paper selects the following input–output variables, which are described in Table 1.

**Table 1.** Selection of input–output variables of carbon dioxide emission efficiency.

| Input/Output | Variable | Descriptive Variable | Data Source |
|---|---|---|---|
| Input variable | Labor force | Number of the employed population at the end of each city (unit: 10,000 people) | Mainly from EPS data platform, China Economic and Social Big Data Research Platform, China Energy Statistics Yearbook, and China Urban Statistics Yearbook. |
| | Capital stock | The stock of fixed assets (unit: 100 million yuan) | |
| | Total energy consumption | Total energy consumption of cities over the years (unit: 10,000 tons of standard coal) | |
| Expected output | Gross Regional Product | Gross domestic product of each urban area over the years (unit: 100 million yuan) | |
| Unexpected output | Carbon dioxide emissions | Carbon dioxide emissions by cities (unit: ten thousand tons). According to the calculation formula in the Guidelines for National Greenhouse Gas Inventories compiled by the United Nations Intergovernmental Panel on Climate Change in 2006 (IPCC, 2006) | |

### 3.3. The Efficiency of Carbon Dioxide Emission in Three Major Urban Agglomerations

Based on the urban data of three major urban agglomerations in China from 2006 to 2019, this paper constructs the SBM-DEA model to calculate each city's carbon dioxide emission efficiency. From Figure 2, it can be seen that the urban carbon dioxide emission efficiency of the three major urban agglomerations is on the rise. In 2006, the overall emission efficiency was about 0.6500, and in 2019, it was about 0.8300, an increase of about 23%. The Pearl River Delta has the highest carbon emission efficiency among the three urban agglomerations. From 2006 to 2019, the average carbon emission efficiency was 0.8345. The carbon emission efficiency of the Yangtze River Delta region followed, with an average of 0.7909. The Beijing–Tianjin–Hebei area has the lowest carbon emission efficiency, with an average of 0.6124.

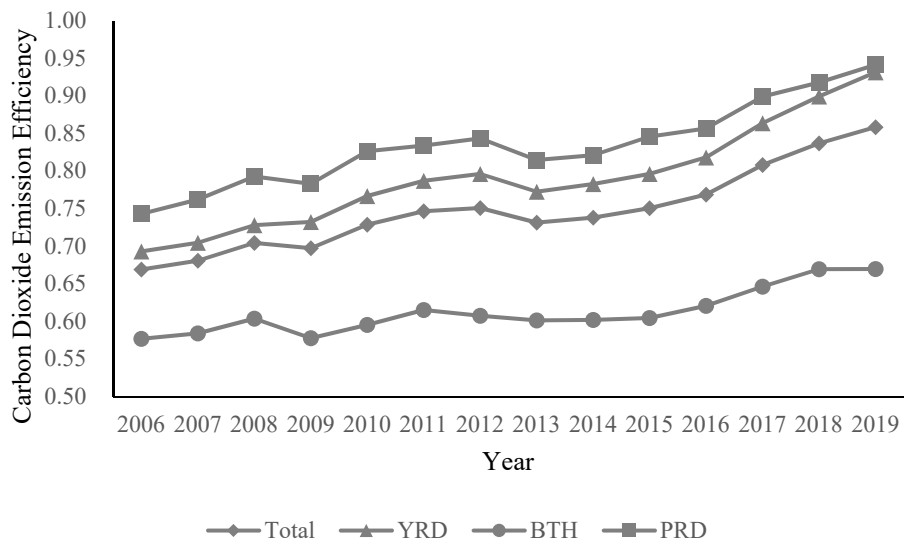

**Figure 2.** $CO_2$ emission efficiency of China's three major urban agglomerations from 2006 to 2019.

## 4. Model Construction and Data Description

### 4.1. Econometric Model

This paper first uses the panel data of Beijing–Tianjin–Hebei, Yangtze River Delta, and Pearl River Delta from 2006 to 2019 to build DID models to analyze the impact of high-speed rail construction on carbon dioxide emission efficiency. According to the first law of geography, everything is connected—as the distance is closer, the connection is more potent [48]. Thus, this paper intends to add spatial factors into the benchmark model to test whether there is a spatial effect. The specific model-building process is as follows, and Figure 3 presents the logic of model construction.

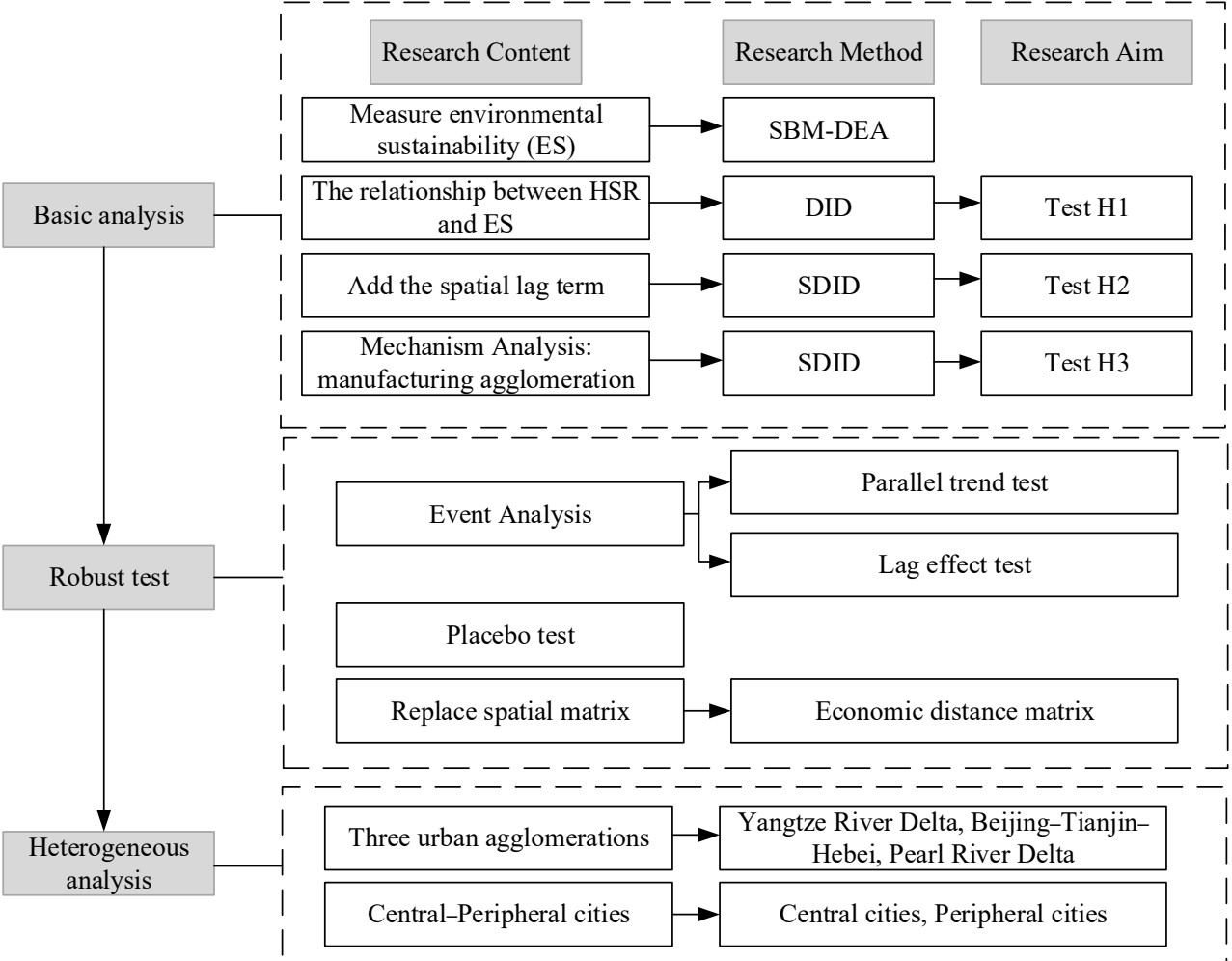

**Figure 3.** The logic roadmap of the model construction.

### 4.1.1. Benchmark Regression: Multiple DID Model

We will build the following econometric models for empirical analysis to verify the above assumptions. Firstly, we take urban agglomeration's carbon dioxide emission efficiency as the explained variable. The construction of high-speed rail is the core explanatory variable. Regional development indicators such as per capita GDP and capita financial expenditure are the control variables. According to the above analysis, we referred to the methodology of Zheng [49] and constructed the benchmark measurement model as follows:

$$ES_{it} = \alpha + \beta_1 HSR + X'_{it}\gamma + \varphi + \tau + \varepsilon_{it} \tag{2}$$

where the subscripts $i$ and $t$ refer to the $i$-th city and $t$-th year, $ES_{it}$ is the indicator of carbon dioxide emission efficiency level in the city $i$ during year $t$, $HSR$ represents the

opening of high-speed rail (the opening record is 1, and the non-opening record is 0), $X_{it}$ is a vector of the control variable (including per capita GDP, per capita financial expenditure, etc.), $\varphi$ is a time-invariant regional fixed effect (e.g., climate and topography, not all of which are observed), $\tau$ is expressed as a time trend effect, which captures unobserved country-wide shocks in any given year that could affect carbon emission dioxide efficiency, $\varepsilon$ is an independent and identically distributed random error term, and $\alpha$, $\beta$, and $\gamma$ are the coefficients that need to be estimated.

### 4.1.2. Spatial Econometric Model: SDID Model

(1) Spatial Autocorrelation Test

To judge whether there is a spatial correlation in carbon dioxide emission efficiency of urban agglomerations, the global Moran's I index is usually used; the formula of this index is as follows:

$$I = \frac{1}{\sum\limits_{i=1}^{n}\sum\limits_{j=1}^{n} w_{ij}} = \frac{\sum\limits_{i=1}^{n}\sum\limits_{j=1}^{n} w_{ij}(x_i - \overline{x})(x_j - \overline{x})}{\sum\limits_{i=1}^{n}(x_i - \overline{x})^2 / n} \tag{3}$$

where $x_i$ and $x_j$ represent the observed values of regions $i$ and $j$, respectively (in this paper, the observed values are the carbon dioxide emission efficiency) and $w_{ij}$ is the spatial weight matrix. The regions positively correlate if the Moran I value is greater than zero. Otherwise, there is a negative correlation. When the Moran I value equals zero, there is no spatial correlation between the areas.

In the above model, $w_{ij}$ a spatial weight matrix is used to express the degree of inter-regional connection, and it is the focus of the spatial econometric model. This paper constructs two matrices: geographical distance matrix and economic distance matrix. These three spatial weight matrices represent the geographical and economic relations between regions, which are constructed explicitly as follows:

① Geographical distance matrix

If the relative size of the distance is considered, spatial adjacency can be described from a quantitative perspective, and the weights defined as:

$$W_{ij}^{dis} = \begin{cases} 1/d_{ij}^2, & \text{if } i \neq j \\ 0, & \text{if } i = j \end{cases} \tag{4}$$

$$d_{ij} = ar\cos\left[\left(\sin\phi_i \times \sin\phi_j\right) + \left(\cos\phi_i \times \cos\phi_j \times \cos(\Delta\tau)\right)\right] \times R \tag{5}$$

where $\phi_i$ and $\phi_j$ are the latitude and longitude of a specific city, respectively, $\Delta\tau$ is the difference of longitude between two cities, and $R$ is the earth's radius, equal to 3958.761 miles.

② Economic distance matrix

With the rapid development of the transportation industry, geographical space resistance to economic activities is weakened, and the economic distance is more important. Per capita GDP is often used to reflect the economic development of a specific region, and the similarity of per capita GDP indicates that the economic development level of the two places is similar. Therefore, this paper uses the difference of GDP per capita between cities in urban agglomeration from 2006 to 2019 to express the economic distance between regions.

$$W_{ij}^{pgdp} = \begin{cases} 1/|pgdp_i - pgdp_j|, & \text{if } i \neq j \\ 0, & \text{if } i = j \end{cases} \tag{6}$$

(2) Spatial Econometric Model

At present, the literature has built DID or spatial econometric models to explore the impact of high-speed rail construction on urban environmental pollution. However little research has considered DID and spatial factors simultaneously, so the estimated results

may have a certain deviation. In addition, most of the existing literature explored the impact of high-speed rail on the environment from the perspective of all Chinese cities. The influence of each urban agglomeration has not yet been explored. This study intends to build an SDID model to fully explore the impact of high-speed rail construction on urban agglomerations' carbon dioxide emission efficiency. The model is set as follows:

$$ES_{it} = \rho \sum_{j=1}^{N} w_{ij} ES_{it} + \alpha_1 HSR_{it} + Z_{it}\lambda + \gamma T_t + v_i + u_{it} \tag{7}$$

where $\sum_{j=1}^{N} w_{ij} ES_{it}$ is the spatial spillover effect of current carbon dioxide emission efficiency, $w_{ij}$ is the spatial weight matrix, and $Z_{it}$ is the vector of explanatory variables in the model (2).

4.1.3. Mechanism Analysis

This paper adds the interactive term of high-speed rail construction and manufacturing agglomeration to the benchmark model. It intends to explore whether high-speed rail construction affects the efficiency of urban carbon dioxide emissions by promoting manufacturing agglomeration. The specific model is as follows:

$$ES_{it} = \rho \sum_{j=1}^{N} w_{ij} ES_{it} + \alpha_1 HSR_{it} + \alpha_2 maggl_{it} + \alpha_3 HSR_{it} \times maggl_{it} + Z_{it}\lambda + \gamma T_t + v_i + u_{it} \tag{8}$$

where $maggl_{it}$ is the manufacturing agglomeration index in city $i$ in year $t$, $\alpha_3$ is the interaction coefficient between high-speed rail construction and manufacturing agglomeration index. This value significantly indicates that high-speed rail construction affects carbon dioxide emission efficiency by influencing manufacturing agglomeration.

*4.2. Variables Selection*

4.2.1. Explained Variable

This paper intends to explore the impact of high-speed rail construction on environmental sustainability. We have chosen the carbon dioxide emission efficiency to represent it. The calculation of the index has been introduced in Section 3.2.

4.2.2. Core Explanatory Variables

The core variable is the construction of high-speed rail. In this paper, the dummy variable of high-speed rail opening is adopted. The opening record is 1, and the non-opening record is 0. In addition, there are three types of high-speed trains in China, namely, bullet trains (denoted as D), high-speed bullet trains (G), and intercity bullet trains (C).

4.2.3. Mechanism Variable: Manufacturing Industry Agglomeration

At present, the indicators to measure industrial agglomeration include employment density, location entropy index, spatial Gini coefficient, Herfindal index, etc. This paper uses the research of Wang [50] for reference and considers the availability of data and the convenience of calculation. It uses the manufacturing location entropy index to measure the degree of industrial agglomeration in the region. The location entropy index can reflect specific industrial agglomeration development advantages in a particular region. The specific calculation formula is:

$$maggl_{ij} = \frac{L_{ij} / \sum_{j=1}^{m} L_{ij}}{\sum_{i=1}^{n} L_{ij} / \sum_{i=1}^{n} \sum_{j=1}^{m} L_{ij}} \tag{9}$$

where $i$ means different cities in three major urban agglomerations, $j(1, 2, 3, \cdots, m)$ represents the industrial sector, $maggl_{ij}$ represents the agglomeration degree in the industry $j$ in the city $i$, and $L_{ij}$ represents the economic scale of the industry $j$ in the city $i$. Generally speaking, a location entropy index greater than one indicates that the region's manufacturing industry tends to agglomerate. The higher the index, the stronger the degree of agglomeration. In this paper, when calculating the location entropy index of the manufacturing industry, we try to avoid the influence of inflation and price factors on the accuracy of calculation results. The economic scale of the industry is measured by the number of employed people in each city.

### 4.2.4. Control Variables

- Economic level: measured by GDP per capita. Generally speaking, per capita GDP can better reflect the economic development level than GDP, so this paper chooses per capita GDP to control it.
- Per capita financial expenditure: the expenditure in the general budget of local finance/resident population is selected for calculation. Fiscal expenditure is used to provide the residents with public services, which directly influences the residents' quality of life and environment.
- Industrial structure: the ratio of GDP of the service industry to GDP is selected for calculation. The industrial structure is an essential aspect of the supply side, reflecting social and economic development characteristics. The service industry is an important index to reflect the upgrading of industrial structure, so it is closely related to the efficiency of carbon dioxide emission.
- The level of human capital: the proportion of the number of students in the resident population in ordinary colleges and universities is selected. The ratio of efficient students is a dimension of population structure, reflecting the region's innovation potential, so it is essential to improving the efficiency of carbon dioxide emissions.
- Environmental regulation: the green coverage rate of built-up areas is selected as the expression. Environmental regulations directly and significantly impact carbon dioxide emissions, so this paper accordingly controls this variable.

### 4.2.5. Data Description

Considering the opening of China's high-speed rail in 2007, we set the starting time as 2006. In addition, since the current city-level data is updated to 2019, we selected the city-level panel data of three major urban agglomerations from 2006 to 2019. Regional GDP, fiscal expenditure, and other related data come from EPS data platforms and statistical yearbooks and bulletins of various cities. Detailed information about the statistics of variables can be seen in Table 2.

**Table 2.** Descriptive statistics of variables.

| Variable | Symbol | Observation | Average Value | Standard Deviation | Minimum | Maximum |
|---|---|---|---|---|---|---|
| Carbon dioxide emission efficiency | CE | 686 | 0.7479 | 0.1262 | 0.4766 | 1.0000 |
| Opening of high-speed rail | hsr | 686 | 0.4927 | 0.5003 | 0.0000 | 1.0000 |
| Manufacturing agglomeration index | maggl | 686 | 1.5168 | 0.4292 | 0.6333 | 2.4463 |
| Per capita GDP | lnpgdp | 686 | 10.8745 | 0.6521 | 9.0765 | 12.7070 |
| Per capita financial expenditure | lnpfinancial | 686 | 8.7908 | 0.7346 | 6.8668 | 11.0021 |
| Industrial structure | indus | 686 | 0.4404 | 0.1808 | 0.23367 | 4.1702 |
| Human capital level | educ | 686 | 5.1756 | 0.9137 | 2.1973 | 7.1471 |
| Environmental regulation | environ | 686 | 3.7342 | 0.2227 | 2.8166 | 5.9575 |

## 5. Empirical Analysis Results

### 5.1. Benchmark Regression: DID Regression Result

Table 3 shows the DID analysis results of three major urban agglomerations. We can see that the construction of high-speed rail has promoted the efficiency of urban carbon dioxide emissions. However, it has only had a significant positive impact on the Yangtze River Delta region. The effect of high-speed rail construction on carbon dioxide emission efficiency in the Beijing–Tianjin–Hebei area is positive but insignificant. However, the impact of high-speed rail construction on carbon dioxide emission efficiency in the Pearl River Delta region is negative but insignificant.

**Table 3.** DID analysis results of three major urban agglomerations.

| VAR | All Regions | Yangtze River Delta | Beijing–Tianjin–Hebei | Pearl River Delta |
|---|---|---|---|---|
|  | (1) | (2) | (3) | (4) |
| hsr | 0.0242 *** | 0.0157 ** | 0.0169 | −0.0154 |
|  | (0.0076) | (0.0076) | (0.0128) | (0.0163) |
| lnpgdp | 0.2160 *** | 0.1205 * | 0.1920 | 0.2700 *** |
|  | (0.0368) | (0.0593) | (0.1210) | (0.0742) |
| lnfinancial | −0.0701 *** | −0.0103 | −0.0751 | −0.0945 |
|  | (0.0222) | (0.0414) | (0.0557) | (0.0543) |
| indus | 0.0109 | 0.4076 *** | −0.0013 | 0.5200 |
|  | (0.0231) | (0.1072) | (0.0086) | (0.3710) |
| lneduc | −0.0442 ** | −0.0250 | −0.0372 | −0.0258 |
|  | (0.0191) | (0.0301) | (0.0347) | (0.0320) |
| lnenviron | 0.0276** | −0.0430 | 0.0432 | 0.0290 *** |
|  | (0.0132) | (0.0368) | (0.0333) | (0.0054) |
| constant | −0.8800 *** | −0.3333 | −0.7460 | −1.551 ** |
|  | (0.2360) | (0.0333) | (0.7240) | (0.4740) |
| Hausman | 54.8200 *** | 53.7400 *** | 11.4800 * | 7.3400 |
| N | 686 | 364 | 196 | 126 |
| R2 | 0.5690 | 0.6957 | 0.3400 | 0.5940 |

Note: The standard error in brackets, ***, **, * meanss significant at the significance level of 1%, 5%, and 10%, respectively.

From the control variables: (1) The influence of GDP per capita on each region's carbon dioxide emission efficiency is significantly positive, which means that the areas with better economic development are more capable of technological innovation to realize energy saving and emission reduction. (2) The industrial structure has a positive effect on the Yangtze River Delta region, which shows that the continuous upgrading of the industrial structure in this region more effectively promotes the improvement of carbon dioxide emission efficiency. (3) Environmental regulation has a significant positive effect on the Pearl River Delta region, so the region can continue to adopt related environmental regulation policies to promote the local carbon dioxide emission efficiency.

### 5.2. Benchmark Regression: SDID Regression Results

5.2.1. Spatial Correlation Test

This paper calculates the global Moran index of carbon dioxide emission efficiency by taking the geographical distance matrix and the economic distance matrix as the spatial weight matrix to test whether the carbon dioxide emission efficiency has spatial correlation. It can be seen from Table 4 that the Moran index of carbon dioxide emission efficiency is above 0.40, regardless of the geographical distance matrix or the economic distance matrix. That is, there is a high positive correlation. In addition, the index shows a gradual upward trend, which to some extent reflects the increasing relevance among regions. Therefore, the estimation result will be biased if the spatial factor is not considered in the model. Thus, this paper also constructs a spatial DID model to analyze this problem.

**Table 4.** Moran index of carbon dioxide emission efficiency.

| Year | Geographical Distance Matrix | Economic Distance Matrix | Year | Geographical Distance Matrix | Economic Distance Matrix |
|---|---|---|---|---|---|
| 2006 | 0.4120 *** (4.6540) | 0.2570 *** (2.5890) | 2013 | 0.5340 *** (5.7770) | 0.6360 *** (5.9130) |
| 2007 | 0.4500 *** (4.9880) | 0.4060 *** (3.9110) | 2014 | 0.5420 *** (5.8540) | 0.7390 *** (6.8430) |
| 2008 | 0.4970 *** (5.4810) | 0.5270 *** (5.0200) | 2015 | 0.5330 *** (5.7680) | 0.7310 *** (6.7760) |
| 2009 | 0.5490 *** (5.9740) | 0.6110 *** (5.7250) | 2016 | 0.5240 *** (5.6670) | 0.7400 *** (6.8380) |
| 2010 | 0.6220 *** (0.0960) | 0.6650 *** (6.1980) | 2017 | 0.5260 *** (5.6990) | 0.7410 *** (6.8660) |
| 2011 | 0.5400 *** (5.8390) | 0.5740 *** (5.3580) | 2018 | 0.5490 *** (5.9340) | 0.7370 *** (6.8270) |
| 2012 | 0.5650 *** (6.1050) | 0.6200 *** (5.7790) | 2019 | 0.6370 *** (6.8410) | 0.7440 *** (6.8790) |

Note: The standard error in brackets, ***, means significant at the significance level of 1%.

### 5.2.2. SDID Regression Results

From the spatial correlation analysis, it can be found that there is a significant spatial correlation between carbon dioxide emission efficiency in different regions, so this paper added spatial factors into the model to regress. As can be seen from Table 5, considering the spatial factors, the construction of high-speed rail has promoted the urban carbon dioxide emission efficiency of the three major urban agglomerations as a whole. In addition, the overall high-speed rail construction also produced a significant positive spatial spillover effect, which means the high-speed rail construction promoted the efficiency of carbon dioxide emission in its surrounding areas. Compared with the DID result in Table 3, we can find that the coefficient of hsr is smaller. The result represents that the model of DID has exaggerated the effect of hsr on environmental sustainability. The reason is that hsr has also included the impact on the surrounding areas. Thus, the estimation is much more exact when considering the spatial effect.

**Table 5.** SDID analysis results of three major urban agglomerations.

| VAR | All Regions | Yangtze River Delta | Beijing, Tianjin and Hebei | Pearl River Delta |
|---|---|---|---|---|
| | (1) | (2) | (3) | (4) |
| hsr | 0.0112 *** (0.0040) | 0.0009 (0.0050) | 0.0221 *** (0.0086) | −0.0139 (0.0133) |
| lnpgdp | 0.1040 *** (0.0149) | 0.0471 ** (0.0190) | 0.2450 *** (0.0345) | 0.2990 *** (0.0580) |
| lnfinancial | −0.0490 *** (0.0094) | −0.0162 (0.0133) | −0.1100 *** (0.0224) | −0.0952 ** (0.0431) |
| indus | 0.0026 (0.0095) | 0.1270 *** (0.0424) | −0.0070 (0.0092) | 0.5620 *** (0.1440) |
| lneduc | −0.0267 *** (0.0068) | −0.0127 (0.0095) | −0.0367 *** (0.0127) | −0.0389 *** (0.0146) |
| lnenviron | 0.0202 ** (0.0083) | −0.0315 (0.0202) | 0.0630 *** (0.0225) | 0.0242 ** (0.0105) |
| rho | 0.7130 *** (0.0347) | 0.7860 *** (0.0398) | −0.4890 *** (0.1450) | −0.5050 *** (0.1600) |
| lambda | −0.5680 *** (0.0721) | −0.6190 *** (0.0999) | 0.6570 *** (0.0816) | 0.5750 *** (0.1220) |
| Sigma2_e | 0.0014 *** (0.0001) | 0.0011 *** (0.0001) | 0.0012 *** (0.0001) | 0.0017 *** (0.0003) |
| N | 686 | 364 | 196 | 126 |
| R2 | 0.3910 | 0.2450 | 0.2430 | 0.1640 |

Note: The standard error in brackets, ***, ** means significant at the significance level of 1%, 5%, respectively.

However, the impact of high-speed rail construction on the carbon dioxide emission efficiency of the three major urban agglomerations is different. (1) First of all, the high-speed rail construction had no significant effect on the local carbon dioxide emission efficiency in the Yangtze River Delta. However, it has significantly promoted the carbon dioxide emission efficiency in its surrounding areas. The possible reason is that the Yangtze River Delta region has diffused its advanced technology, which led to the improvement of the efficiency of technological innovation in the surrounding areas and then promoted energy saving and emission reduction. (2) Secondly, the construction of high-speed rail has significantly promoted the efficiency of carbon dioxide emission in the Beijing–Tianjin–Hebei region. However, it had a negative spillover effect on surrounding areas, reducing the efficiency of carbon dioxide emission in nearby areas. The possible reason is that high-speed rail construction has caused many resources to flow into the Beijing–Tianjin–Hebei region, thereby promoting technological innovation in the inner region of Beijing–Tianjin–Hebei and realizing low-carbon development. However, due to the outflow of resources from the surrounding areas, they cannot carry out technological innovation, which led to the decline of their carbon dioxide emission efficiency. In addition, the construction of high-speed rail can also lead to the migration of highly polluting manufacturing industries to the surrounding areas, thus reducing their carbon dioxide emission efficiency. (3) Thirdly, the construction of high-speed rail had a negative impact on the carbon dioxide emission efficiency of the Pearl River Delta, but the effect is not significant. At the same time, the construction of high-speed rail also led to the reduction of carbon dioxide emission efficiency in surrounding areas, which is similar to that in the Beijing–Tianjin–Hebei region. Therefore, for these two regions, while improving the construction of high-speed rail networks, we also need to consider its negative effects on the surrounding areas.

Lesage and Pace [51] put forward the concepts and decomposition methods of direct effect, indirect effect, and total effect to solve the problem of uncertain coefficients in the spatial econometric model. Direct effect indicates the influence of independent variables on dependent variables in a particular area, including model coefficient and feedback effect. Feedback effect refers to the impact of independent variables in one region on dependent variables in the other regions. At the same time, other areas, in turn, influence the explained variables in this region. From this point of view, the decomposition of the spatial spillover effect is more accurate and more practical than looking directly at the model coefficients. The indirect effect refers to the influence of local explanatory variables on explained variables in other regions. In contrast, total effect refers to the average impact of local explanatory variables on all areas.

It can be seen from Table 6 that in the overall regression, the direct effect, indirect effect, and total effect of the all-region are significantly positive. The results show that the construction of high-speed rail promotes the efficiency of carbon dioxide emission in this city and brings positive externalities to surrounding cities. However, the Yangtze River Delta and Pearl River Delta coefficients are insignificant. The insignificant results show that the high-speed rail construction has had little impact on these two areas. However, the construction of high-speed rail substantially affected the carbon dioxide emission efficiency in the Beijing–Tianjin–Hebei region. Its total effect is significantly positive, indicating that high-speed rail construction has dramatically improved the carbon dioxide emission efficiency in the Beijing–Tianjin–Hebei area. From the perspective of effect decomposition, the indirect effect of the Beijing–Tianjin–Hebei region is significantly negative. It shows that high-speed rail construction has negative externalities to surrounding areas. To achieve balanced local development, the government needs to pay attention to this negative impact.

**Table 6.** Decomposition of direct effect, indirect effect, and the total effect of three major urban agglomerations.

| Effect | All Regions | Yangtze River Delta | Beijing–Tianjin–Hebei | Pearl River Delta |
|---|---|---|---|---|
| | (1) | (2) | (3) | (4) |
| Direct effect | 0.0139 *** | 0.0014 | 0.0237 ** | −0.0149 |
| | (0.0049) | (0.0067) | (0.0092) | (0.0153) |
| Indirect effect | 0.0258 *** | 0.0033 | −0.0084 ** | 0.0059 |
| | (0.0100) | (0.0188) | (0.0040) | (0.0067) |
| Total effect | 0.0397 *** | 0.0047 | 0.0153 ** | −0.0090 |
| | (0.0145) | (0.0254) | (0.0064) | (0.0093) |

Note: The standard error in brackets, ***, ** means significant at the significance level of 1%, 5%, respectively.

### 5.3. Mechanism Analysis Result

5.3.1. Manufacturing Agglomeration in Three Major Urban Agglomerations

This paper calculates the manufacturing agglomeration degree of each city from 2006 to 2019. Meanwhile, we find significant differences in each urban agglomeration degree's manufacturing agglomeration from Figure 4. On the whole, the degree of manufacturing agglomeration is relatively stable, averaging around 1.0. The Yangtze River Delta urban agglomeration index is also approximately 1.0 on average, declining slightly in 2013. Then it is in a slow upward trend, which indicates that the manufacturing agglomeration in this region is in a continuous development trend. The value of the manufacturing agglomeration index in the Pearl River Delta is above 1.3. The result reflects that the manufacturing agglomeration in this region is in a high development trend—the index has been rising since 2012. However, the agglomeration degree of the manufacturing industry in Beijing–Tianjin–Hebei urban is low, about 0.60, and there is a downward trend after 2012.

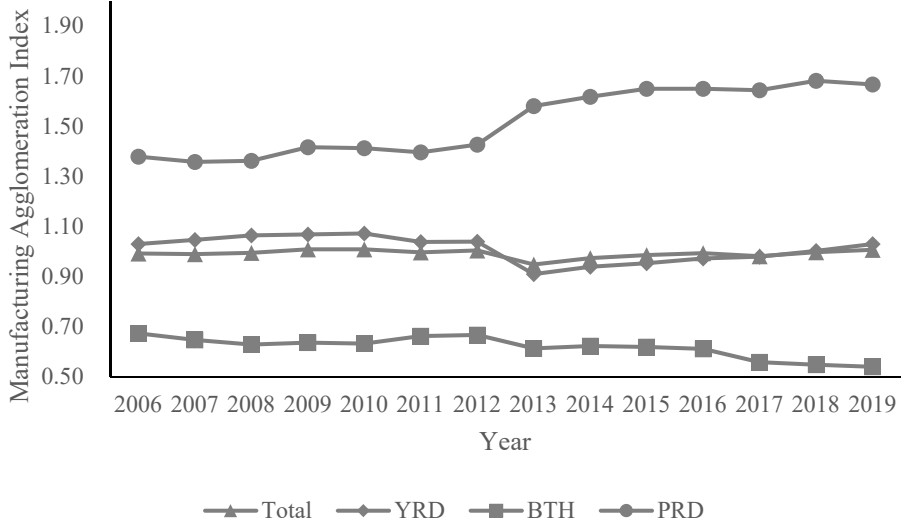

**Figure 4.** Manufacturing agglomeration in three major urban agglomerations from 2006 to 2019.

5.3.2. Mechanism Analysis Results

From the overall regression aspect, the interaction coefficient between high-speed rail construction and manufacturing industry agglomeration is significantly positive, indicating that high-speed rail construction can promote the efficiency of carbon dioxide emission by promoting manufacturing industry agglomeration. In Section 5.3.1, we found that the manufacturing agglomeration of all regions is in a developing trend. Therefore, for the three major urban agglomerations it is necessary to continue to improve the construction of high-speed rail networks to promote the agglomeration of manufacturing industries and exert positive environmental effects.

In Table 7, we can see the analysis results of the mechanism of three major urban agglomerations. As for the Yangtze River Delta region, the interaction coefficient of hsr×maggl is not significant, indicating that the impact of high-speed rail construction on manufacturing industry agglomeration is not significant. However, according to the regression coefficient, the effect of industrial agglomeration on carbon dioxide emission efficiency is significantly positive. The possible reason for the result is that the manufacturing technology level in the Yangtze River Delta region is relatively high, and its agglomeration positively impacts the environment. In the Beijing–Tianjin–Hebei region and the Pearl River Delta region, manufacturing industry agglomeration has had a negative impact on carbon dioxide emission efficiency, which indicates that manufacturing industries in these two regions are highly polluting enterprises. However, high-speed rail construction has not promoted the industrial agglomeration in the Beijing–Tianjin–Hebei region, negatively impacting the environment. In recent years, the manufacturing agglomeration degree in this area has been low and has a downward trend, so this mechanism does not exist. However, in the Pearl River Delta region, the construction of high-speed rail significantly has a negative impact on the environment by promoting the agglomeration of manufacturing industries. From the previous analysis, it can also be seen that the manufacturing industry agglomeration in the Pearl River Delta region is on the rise. Hence, the area needs to pay attention to its negative impact when vigorously developing high-speed rail construction.

**Table 7.** Analysis results of the mechanism of three major urban agglomerations.

| VAR | All Regions | Yangtze River Delta | Beijing–Tianjin–Hebei | Pearl River Delta |
|---|---|---|---|---|
| | (1) | (2) | (3) | (4) |
| hsr | −0.0016 | −0.0087 | 0.0443 ** | 0.1486 *** |
| | (0.0071) | (0.0100) | (0.0206) | (0.0360) |
| maggl | −0.0098 | 0.0407 *** | −0.0642 * | −0.0438 * |
| | (0.0094) | (0.0114) | (0.0360) | (0.0253) |
| hsr × maggl | 0.0126 ** | 0.0096 | −0.0294 | −0.0964 *** |
| | (0.0057) | (0.0085) | (0.0283) | (0.0231) |
| lnpgdp | 0.0983 *** | 0.0361 * | 0.2373 *** | 0.1576 *** |
| | (0.0150) | (0.0196) | (0.0352) | (0.0535) |
| lnfinancial | −0.0455 *** | −0.0032 | −0.1067 *** | −0.0153 |
| | (0.0096) | (0.0196) | (0.0222) | (0.0386) |
| indus | 0.0041 | 0.1582 *** | −0.0071 | 0.2260 * |
| | (0.0094) | (0.0454) | (0.0092) | (0.1278) |
| lneduc | −0.0280 *** | −0.0163 * | −0.0325 ** | −0.0157 |
| | (0.0071) | (0.0097) | (0.0129) | (0.0139) |
| lnenviron | 0.0211 ** | −0.0457 ** | 0.0487 ** | −0.0020 |
| | (0.0086) | (0.0212) | (0.0233) | (0.0125) |
| rho | 0.7155 *** | 0.7540 *** | −0.4179 *** | 0.1299 |
| | (0.0343) | (0.0460) | (0.1551) | (0.1882) |
| lambda | −0.5887 *** | −0.5306 *** | 0.6165 *** | 0.2518 |
| | (0.0086) | (0.1118) | (0.0905) | (0.1980) |
| Sigma2_e | 0.0014 *** | 0.0010 *** | 0.0012 *** | 0.0013 *** |
| | (0.0001) | (0.0001) | (0.0001) | (0.0002) |
| N | 686 | 364 | 196 | 126 |
| R2 | 0.5602 | 0.7380 | 0.3779 | 0.7133 |

Note: The standard error in brackets, ***, **, * means significant at the significance level of 1%, 5%, and 10%, respectively.

### 5.4. Robustness Test

5.4.1. Event Analysis Results

The parallel trend of carbon dioxide emission efficiency in cities with and without high-speed rail is essential for double differential estimation. To test the parallel trend, refer

to Xu [52] and add the dummy variables of the front and back terms of high-speed rail connection based on Equation (2):

$$CE_{it} = \sum_{m=1}^{4} \lambda_m FirstHSR_{i,t-m} + \sum_{n=0}^{3} \lambda_n FirstHSR_{i,t+n} + X'_{it}\gamma + \varphi + \tau + \varepsilon_{it} \quad (10)$$

where $FirstHSR_{it}$ is a dummy variable representing the first time to open high-speed rail in the city $i$ at $t$, $FirstHSR_{i,t-m}$ means the preceding item of the $m$ period, and $FirstHSR_{i,t+n}$ defines the lag term of the $n$-th period. The former tests the effect before the opening of high-speed rail and verifies the parallel hypothesis. The lag term is used to identify the impact after the opening of the high-speed rail. From Table 8 and Figure 5, we could find that the former coefficient of high-speed rail construction is insignificant, indicating a parallel trend between the experimental group and the control group before opening the high-speed rail. In addition, as shown from Figure 5, there is a significant impact on carbon dioxide emission efficiency in the third year after the opening of high-speed rail, which indicates that the construction effect of high-speed rail has a certain lag.

**Table 8.** Robustness Test (Event Analysis Results).

| VAR | DID | SDID |
|---|---|---|
| hsr (−4) | −0.0055 (0.0084) | −0.0066 (0.0060) |
| hsr (−3) | −0.0108 (0.0072) | −0.0082 (0.0068) |
| hsr (−2) | 0.0069 (0.0043) | 0.0033 (0.0068) |
| hsr (−1) | 0.0156 *** (0.0052) | 0.0092 * (0.0056) |
| hsr (1) | −0.0006 (0.0054) | −0.0009 (0.0056) |
| hsr (2) | 0.0024 (0.0058) | −0.0007 (0.0065) |
| hsr (3) | 0.0408 *** (0.0060) | 0.0224 *** (0.0057) |
| lnpgdp | 0.2179 *** (0.0362) | 0.1187 *** (0.0155) |
| lnfinancial | −0.0820 *** (0.0217) | −0.0584 *** (0.0099) |
| indus | −0.0004 (0.0173) | −0.0059 (0.0095) |
| lneduc | −0.0408 ** (0.0180) | −0.0256 *** (0.0068) |
| lnenviron | 0.0312 ** (0.0150) | 0.0272 *** (0.0083) |
| cons | −0.8204 *** (0.2429) | |
| rho | | 0.6645 *** (0.0394) |
| lambda | | −0.5247 *** (0.0776) |
| Sigma2_e | | 0.0014 *** (0.0001) |
| N | 686 | 686 |
| R² | 0.6087 | 0.6100 |

Note: The standard error in brackets, ***, **, * means significant at the significance level of 1%, 5%, and 10%, respectively.

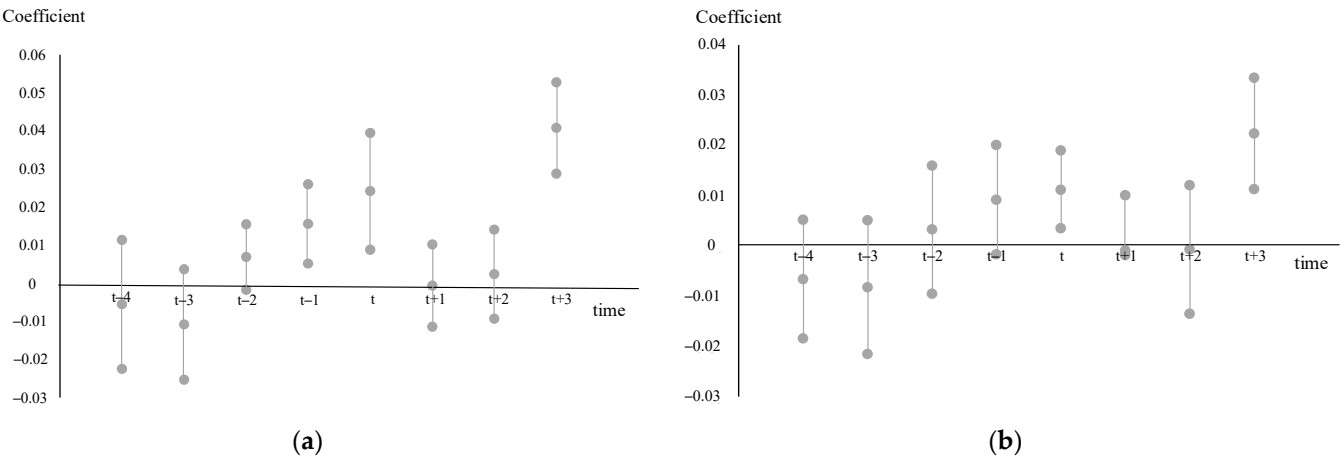

**Figure 5.** Event analysis results: (**a**) DID event analysis results; (**b**) SDID event analysis results.

### 5.4.2. Placebo Test

This paper also adopted the placebo test to ensure the robustness of the research results. The specific processes are as follows: (1) Firstly, randomly generate 1000 high-speed rail variables $HSR^{false}$. (2) Secondly, replace the original variable with the newly generated high-speed rail variable $HSR$. (3) Finally, using 1000 new samples, we constructed DID and SDID models to estimate the coefficients. We can judge that benchmark regression results are not accidental by constructing random samples. In other words, the influence is real. From Figure 6, we can find that the true coefficient is at the right end of the nuclear density curve. The result shows that this event is a low probability event, which means the impact of high-speed rail construction on carbon dioxide emission efficiency exists.

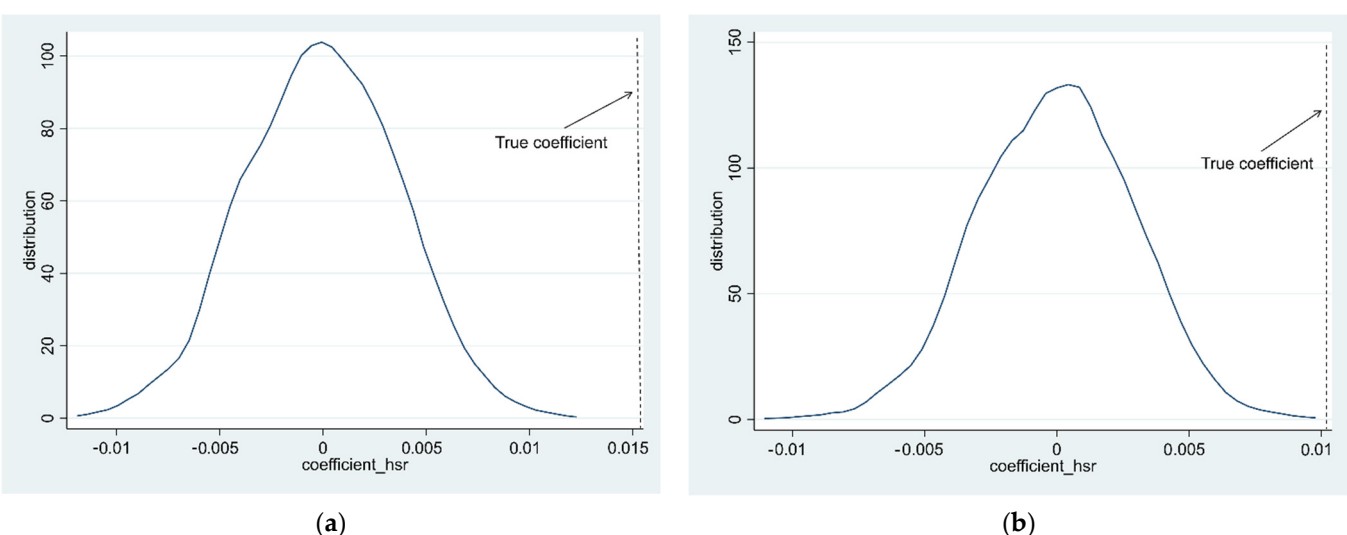

**Figure 6.** Placebo tests: (**a**) DID placebo test; (**b**) SDID placebo test.

### 5.4.3. Endogenous Test: Instrumental Variable Method

Because high-speed rail construction is non-random, this variable has some endogenous problems. Studies have pointed out that it is often easier to plan and build high-speed rail in areas with better economic development [53]. This paper further uses instrumental variables to solve this endogenous problem. Based on the research of Wang [17], we intend to use the average slope of each city calculated based on Arc GIS to construct the instrumental variable of high-speed rail opening. Geographical slope can effectively reflect the change of terrain and then effectively measure the construction difficulty of high-speed rail. The larger the average urban slope, the more difficult it is to build high-speed rail, so there is a

correlation between the geographical slope and the opening of high-speed rail. The slope is a natural geographical condition formed by long-term history, and it is not directly related to other economic indicators which meet the exogenous requirements of instrumental variables. From Table 9, we could find that the p-values of all Sargan tests are significantly greater than 0.05, indicating that there is no over-recognition. In addition, in the regression results, the variable of high-speed rail opening is significantly positive. The result shows that the construction of high-speed rail has dramatically promoted the efficiency of carbon dioxide emission in urban agglomerations, and the above results are stable.

**Table 9.** Robustness test (instrumental variable method).

| VAR | DID + IV | SDID + IV |
|---|---|---|
| hsr | 0.0373 *** | 0.0328 *** |
| | (0.0090) | (0.0089) |
| lnpgdp | 0.2014 *** | 0.2077 *** |
| | (0.0206) | (0.0201) |
| lnfinancial | −0.0667 *** | −0.0720 *** |
| | (0.0145) | (0.0142) |
| indus | 0.0031 | 0.0027 |
| | (0.0121) | (0.0118) |
| lneduc | −0.0367 *** | −0.0324 *** |
| | (0.0094) | (0.0092) |
| lnenviron | 0.0416 *** | 0.0372 *** |
| | (0.0107) | (0.0105) |
| cons | −0.8419 *** | |
| | (0.1290) | |
| rho | | 0.3944 *** |
| | | (0.0803) |
| Hausman | 72.6800 *** | 102.1300 *** |
| Sargan | 0.8920 | 3.4100 |
| p | 0.3448 | 0.0548 |
| N | 637 | 637 |
| $R^2$ | 0.5592 | 0.5785 |

Note: The standard error in brackets, *** means significant at the significance level of 1%.

### 5.4.4. Replace the Spatial Weight Matrix

In addition, this paper also changed the spatial weight matrix to test its robustness. Table 10 shows the coefficient of regression result changed to the economic distance matrix. It can be seen from the table that the coefficient of high-speed rail construction is positive, which indicates that the high-speed rail construction has significantly promoted the efficiency of carbon dioxide emission in urban agglomerations, and the research results are robust.

**Table 10.** Robustness test (replacing spatial weight matrix).

| VAR | All Regions | Yangtze River Delta | Beijing–Tianjin–Hebei | Pearl River Delta |
|---|---|---|---|---|
| | (1) | (2) | (3) | (4) |
| hsr | 0.0125 *** | 0.0008 | 0.0137 * | −0.0104 |
| | (0.0044) | (0.0047) | (0.0082) | (0.0132) |
| lnpgdp | 0.1210 *** | 0.0444 ** | 0.2190 *** | 0.2190 *** |
| | (0.0174) | (0.0176) | (0.0324) | (0.0563) |
| lnfinancial | −0.0490 *** | −0.0059 | −0.1010 *** | −0.0899 ** |
| | (0.0106) | (0.0120) | (0.0209) | (0.0397) |

**Table 10.** *Cont.*

| VAR | All Regions | Yangtze River Delta | Beijing–Tianjin–Hebei | Pearl River Delta |
|---|---|---|---|---|
| | (1) | (2) | (3) | (4) |
| indus | 0.0058 | 0.0910 ** | −0.0014 | 0.3510 ** |
| | (0.0094) | (0.0395) | (0.0084) | (0.1400) |
| lneduc | −0.0282 *** | −0.0188 ** | −0.0142 | −0.0311 * |
| | (0.0069) | (0.0088) | (0.0125) | (0.0159) |
| lnenviron | 0.0249 *** | −0.0327 * | 0.0662 *** | 0.0291 ** |
| | (0.0085) | (0.0183) | (0.0215) | (0.0115) |
| rho | 0.6160 *** | 0.7590 *** | 0.1680 | 0.5080 *** |
| | (0.0450) | (0.0354) | (0.1340) | (0.1070) |
| lambda | −0.3130 *** | −0.7130 *** | 0.4380 *** | −0.0907 |
| | (0.0800) | (0.0595) | (0.1190) | (0.1970) |
| Sigma2_e | 0.0015 *** | 0.0010 *** | 0.0011 *** | 0.0018 *** |
| | (0.0001) | (0.0001) | (0.0001) | (0.0002) |
| N | 686 | 364 | 196 | 126 |
| R2 | 0.4520 | 0.3020 | 0.3460 | 0.3590 |

Note: The standard error in brackets, ***, **, * means significant at the significance level of 1%, 5%, and 10%, respectively.

*5.5. Further Analysis: Central Cities and Peripheral Cities*

The capital attraction is significantly different because central and peripheral cities have different market sizes and resource endowments. Therefore, the impact of high-speed rail construction on the two types of cities may differ. Therefore, this paper analyzes the effects of high-speed rail construction on central and peripheral cities' carbon dioxide emission efficiency. In this paper, municipalities, provincial capitals, and sub-provincial cities are divided into central cities, and other cities are peripheral cities. The central cities of the three major urban agglomerations are Beijing, Shanghai, Tianjin, Chongqing, Nanjing, Hefei, Shijiazhuang, Guangzhou, Hangzhou, Shenzhen, and Ningbo.

The results in Table 11 show that high-speed rail construction has no significant impact on central cities' carbon dioxide emission efficiency. However, it has improved the carbon emission efficiency of peripheral cities. The possible reason for this is that the high-speed rail construction will promote a large amount of highly skilled labor and technology to flow into the peripheral cities. This will significantly improve the technical level of the highly polluting manufacturing industry, thus facilitating its carbon emission efficiency. As for the central cities, the local manufacturing enterprises' technical level is high, so the high-speed rail construction has no significant impact. In addition, although the construction of high-speed rail does not promote the carbon emission efficiency of central cities, its spatial spillover effect is significantly positive. The results indicate that central cities will positively affect surrounding cities and promote technological innovation, thereby improving the carbon dioxide emission efficiency. Therefore, urban agglomerations should continue to strengthen the construction of a high-speed rail network, accelerate the connection between central cities and peripheral cities, and promote the flow of advanced technology into peripheral cities to realize the green development of the whole region.

**Table 11.** Regression results of central cities and peripheral cities.

| VAR | Central City | | Peripheral City | |
|---|---|---|---|---|
| | DID | SDID | DID | SDID |
| hsr | 0.0160 | 0.0057 | 0.0227 *** | 0.0111 *** |
| | (0.0153) | (0.0062) | (0.0078) | (0.0041) |
| lnpgdp | 0.1997 ** | 0.1029 *** | 0.1929 *** | 0.0883 *** |
| | (0.0854) | (0.0239) | (0.0398) | (0.0157) |

**Table 11.** *Cont.*

| VAR | Central City | | Peripheral City | |
|---|---|---|---|---|
| | **DID** | **SDID** | **DID** | **SDID** |
| lnfinancial | −0.0370 | −0.0167 | −0.0685 *** | −0.0462 *** |
| | (0.0587) | (0.0186) | (0.0229) | (0.0097) |
| indus | 0.3860 | 0.0815 | 0.0020 | −0.0023 |
| | (0.2121) | (0.0646) | (0.0147) | (0.0087) |
| lneduc | −0.0631 | −0.0984 *** | −0.0259 | −0.0066 |
| | (0.0348) | (0.0209) | (0.0189) | (0.0066) |
| lnenviron | 0.1200 | 0.0047 | 0.0265 * | 0.0198 ** |
| | (0.0947) | (0.0329) | (0.1377) | (0.0077) |
| constant | −1.4572 *** | | −0.7166 *** | |
| | (0.3991) | | (0.2440) | |
| rho | | 0.6109 *** | | 0.7220 *** |
| | | (0.0621) | | (0.0402) |
| lambda | | −0.6122 *** | | −0.3881 *** |
| | | (0.0791) | | (0.0936) |
| Sigma2_e | | 0.0007 *** | | 0.0011 *** |
| | | (0.0001) | | (0.0001) |
| N | 140 | 140 | 546 | 546 |
| R2 | 0.8355 | 0.8786 | 0.5032 | 0.5319 |

Note: The standard error in brackets, ***, **, * means significant at the significance level of 1%, 5%, and 10%, respectively.

## 6. Conclusions and Policy Implications

This paper calculated each city's carbon dioxide emission efficiency by SBM-DEA model based on the panel data of three major urban agglomerations in China from 2006 to 2019. Then, we constructed DID and spatial DID models to explore the impact of high-speed rail construction on the sustainable urban environment of the three major urban agglomerations. The research found that:

On the whole, the construction of high-speed rail has significantly improved the carbon dioxide emission efficiency in local and surrounding urban areas, which is conducive to the sustainable development of the local environment. In addition, from the impact of high-speed rail construction on the carbon dioxide emission efficiency of central cities and peripheral cities, we can see that the opening of high-speed rail has no significant effect on the carbon emission efficiency of central cities. However, it has significantly promoted all regions' positive spatial spillover effect. In other words, it has enabled the green and low-carbon development of neighboring areas.

Regarding regions, the impact of high-speed rail construction on three major urban agglomerations' carbon dioxide emission efficiency is different.

First of all, the construction of high-speed rail has significantly promoted the efficiency of carbon dioxide emission in the Beijing–Tianjin–Hebei region. However, it has had a negative spatial spillover effect on neighboring areas. The possible reason is that high-speed rail construction promotes the loss of many resources around Beijing, Tianjin, and Hebei. As a result, the resource allocation efficiency is not high, which further inhibits carbon dioxide emission efficiency from improving.

Secondly, the construction of high-speed rail has significantly promoted the spatial agglomeration of the manufacturing industry in the Pearl River Delta region. However, industrial agglomeration has brought about the reduction of local carbon dioxide emission efficiency. The possible reason is that high-speed rail construction stimulated the agglomeration of the highly polluting manufacturers, thereby inhibiting regional carbon dioxide emission efficiency.

Thirdly, considering the spatial factors, the impact of high-speed rail construction on the carbon emission efficiency in the Yangtze River Delta region is not apparent. The possible reason may be the high level of technology in the Yangtze River Delta region itself. The impact of high-speed rail construction on the innovation level of its manufacturing

industry is not significant. Thus the effect on the local carbon emission efficiency is not significant.

From the event analysis results, the impact of high-speed rail construction on urban agglomerations' urban carbon dioxide emission efficiency has a lagging effect. From the regression results, the high-speed rail construction has the most significant impact on environmental efficiency when it lags for three periods. The reason is that the construction of high-speed rail first promotes the free flow of factors, but its influence on factor allocation efficiency will take some time.

The above conclusions have the following policy implications for the sustainable development of China's urban agglomeration environment:

Firstly, the Chinese government should continue to improve the construction of high-speed rail networks in urban agglomerations to promote the sustainable development of the environment. In the future, the Chinese government needs to continuously strengthen the construction of high-speed rail to realize the following development goals: 1–3 h traffic circle between neighboring large and medium-sized cities and 0.5–2 h traffic circle within urban agglomerations. The high-speed rail will exert the space–time compression effect, promoting the efficiency of factor space allocation and enhancing the city's innovation ability, leading to the improvement of the carbon dioxide emission efficiency.

Secondly, each urban agglomeration should take heterogeneous measures to realize environmental sustainability according to its characteristics. For the Beijing–Tianjin–Hebei region, the local government should strengthen the links between central and peripheral cities and enhance their cooperation to promote technological innovation in the peripheral areas, improving carbon dioxide emission efficiency. For the Pearl River Delta region, the local government needs to adopt strategies to encourage local high-pollution manufacturing industries to innovate for saving energy and reducing emissions. For instance, they could give them more energy-saving and emission-reduction subsidies to achieve green and low-carbon development. For the Yangtze River Delta region, we will continue to upgrade the local industrial structure and promote the agglomeration of manufacturing industries to realize the high-quality development of the area.

Finally, the construction of high-speed rail needs a long-term evaluation from the whole life cycle perspective to maximize cost savings and improve the social effects of transportation infrastructure. The lagging impact of high-speed rail construction on the environment exists, so the government needs to evaluate the short-term and long-term environmental costs and benefits of high-speed rail construction in advance. Then, we could determine the quantity and quality of high-speed rail construction in this area to maximize overall social welfare.

**Author Contributions:** Conceptualization, S.Z. and Z.Z.; methodology, R.L.; software, S.Z.; validation, S.Z., Z.Z. and R.L.; formal analysis, S.Z.; investigation, S.Z.; resources, Z.Z.; data curation, W.L.; writing—original draft preparation, S.Z.; writing—review and editing, S.Z. and Z.Z.; visualization, W.L.; supervision, R.L.; project administration, R.L.; funding acquisition, W.L. All authors have read and agreed to the published version of the manuscript.

**Funding:** This research was funded by National Railway Administration (Grant No. B19DJ00030), China.

**Institutional Review Board Statement:** Not applicable.

**Informed Consent Statement:** Not applicable.

**Data Availability Statement:** The data supporting this study's findings are available in the "China Statistical Yearbook" and provincial and municipal statistical yearbooks. These data were derived from the following resources available in the public domain: http://www.stats.gov.cn/tjsj/ndsj/ (accessed on 22 January 2022).

**Conflicts of Interest:** The authors declare no conflict of interest.

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
