# Peer review of "Impact of High-Speed Rail Construction on the Environmental Sustainability of China’s Three Major Urban Agglomerations"

_sustainability, doi:10.3390/su14052567_

Round 1
Reviewer 1 Report
In this paper, the authors use the city-level panel data of the three urban agglomerations from 2006 to 2019 to construct the SBM-DEA model for calculating each city's carbon dioxide emission efficiency. The conclusions are helpful for people to objectively understand the impact of high-speed rail construction on each urban agglomeration's carbon dioxide emission efficiency, and promote the sustainable development. The manuscript is well organized, and the results enrich the existing theoretical research models and have certain practical meaning. Therefore, I think the paper can be accepted. In addition, the following are the few comments, which may be included while revision.
- All of the abbreviations in the text should give their full name when they first appears, such as CCR, BCC, F.G., S.T., and SBM on page 6.
- There are many equations used in the paper and the physical or realistic meaning of all the variables in these equations should be given, such as in Equation (1).
- It is recommended that the authors give a detailed explanation of the objective function and each constraint in Equation 1 to facilitate the reader's understanding.
- The four polylines in Figures 3 and 4 are difficult to distinguish when the manuscript is printed in black and white color. It is recommended that the author distinguish these different polylines with small triangles or small squares to make them easier for the reader to read.
- Figure 1, Figure 2, Figure 4, Figure 6, Table 2, Table 7, and Table 11 are not cited in the manuscript.
- Why use SBM-DEA method to study this problem? Different methods often lead to different results, so what are the advantages of this method? This should be given in the part of methodology.
- It is suggested that the author compare the research results in this paper with the existing models, so that readers can easily find the advantages of the method used in paper.
- The list of references should be extended to include some recent papers as follow.
- Research on the Influence of a High-Speed Railway on the Spatial Structure of the Western Urban Agglomeration Based on Fractal Theory—Taking the Chengdu–Chongqing Urban Agglomeration as an Example. Sustainability 2020, 12(18), 7550; https://doi.org/10.3390/su12187550.
- Impacts of high-speed railway on the industrial pollution emissions in China: Evidence from multi-period difference-in-differences models. Kybernetes, 2020.
- How Does Transportation Infrastructure Improve Corporate Social Responsibility? Evidence from High-Speed Railway Openings in China[J]. Sustainability, 2021, 13(11):6455.
Reviewer 2 Report
The work is interesting because it proposes a method of calculating the impact on the environment of the construction of high-speed railways. The method is applied to three large urban agglomerations in China. It considers both the positive and negative effects in the area affected by the construction and the spatial spillover.
Some comments are reported below
- Model construction evaluation (section 4) requires specific expertise on econometric models. To facilitate understanding even by non-specialists, it would be useful to report the correspondence between the symbols shown in the formulas and the variables and constants of the cases examined: Formula (1) line 258, symbols s, y, b, z, N, M, I ; Formula (2) row 297 symbols X, Ï•, t; Formula (3) row 308 symbols X, w, n; variables of the case examined; Formula (9) line 377 symbol L;
- Section 5.3.2. Prove the statement reported in line 533-535: which table and which variable are you referring to?
- Check the numbering of the figures: figure 2 is shown after figure 3
- Report in figures 3, 4, 5, 6 the title and unit of measurement of the axes of the diagrams.
